# Assessment of Inspiratory Muscle Function and Glenohumeral Motion in the Throwing Arm of Division I Collegiate Baseball Players

**Luis A. Feigenbaum** [1,2,*], **Julian J. Rivera** [1,2], **Michele A. Raya** [1], **Meryl I. Cohen** [1], **Lee D. Kaplan** [3] **and Lawrence P. Cahalin** [1]

1   Department of Physical Therapy, Miller School of Medicine, University of Miami, Coral Gables, FL 33146, USA
2   Department of Intercollegiate Athletics, University of Miami, Coral Gables, FL 33146, USA
3   UHealth Sports Medicine Institute, University of Miami, Miami, FL 33136, USA
*   Correspondence: lfeigenbaum@miami.edu

**Abstract**

This study investigated the relationships between inspiratory performance (IP) and glenohumeral rotation in Division 1 Collegiate baseball players (D1CBP). Thirty D1CBP were recruited. The Test of Incremental Respiratory Endurance (TIRE) provides maximal inspiratory pressure (MIP), sustained maximal inspiratory pressure (SMIP), and inspiratory duration (ID). Right and left glenohumeral internal and external rotation (RGHIR, RGHER, LGHIR, and LGHER, respectively) were measured with the shoulder in 90 degrees(d) of abduction. Significant differences between position groups were observed. IP of the entire group was significantly correlated to height, weight, and negatively correlated to right total rotational motion (RTRM) (r = −0.41; $p < 0.05$). The IP of all pitchers was significantly negatively correlated to both RTRM and LTRM (r = −0.56 to −0.61; $p < 0.05$). IP of right-handed pitchers was significantly correlated negatively to RGHER (r = −0.83 to −0.93; $p < 0.05$). IP of left-handed pitchers was significantly correlated negatively to LGHER (r = −0.82; $p = 0.04$). GH motions are significantly related to the IP of D1CBP. This association may be explained by the involvement of overstretched internal rotators, which act as accessory inspiratory muscles.

**Keywords:** inspiratory muscle performance; inspiratory muscle test; baseball; glenohumeral rotation; total rotational shoulder motion

## 1. Introduction

Throwing a baseball is a dynamic, whole-body movement that places significant biomechanical and physiological demands on the athlete, particularly on the throwing arm and respiratory system [1]. The repetitive, high-intensity nature of throwing in Division I collegiate baseball players (D1CBP) can lead to adaptations in the shoulder's glenohumeral joint and inspiratory muscle function, which are critical for performance and injury prevention [2,3]. Understanding these adaptations is essential, as shoulder injuries and fatigue-related performance declines are prevalent in overhead throwing sports, affecting athletes' careers and team outcomes [4,5].

The glenohumeral joint in baseball pitchers often exhibits unique range of motion (ROM) characteristics, including increased external rotation (ER) and decreased internal rotation (IR), a phenomenon known as glenohumeral internal rotation deficit (GIRD) [6,7].

These adaptations are attributed to repetitive throwing, which induces osseous changes, such as humeral retroversion, and soft tissue alterations, including posterior capsule thickening [8,9]. GIRD has been identified as a risk factor for shoulder injuries, such as internal impingement and rotator cuff tears, prompting research into its biomechanical underpinnings [10]. Recent studies have also linked lower extremity flexibility and trunk mechanics to glenohumeral ROM, suggesting that pitching is a kinetic chain activity requiring coordinated movement across multiple body segments [11,12]. However, controversy exists regarding whether GIRD is a pathological condition or a normal adaptation in asymptomatic pitchers, with some studies advocating for targeted stretching to mitigate deficits, while others suggest that excessive ROM restoration may destabilize the shoulder [13,14].

Concurrently, inspiratory muscle function has emerged as a critical factor in athletic performance, particularly in sports requiring sustained high-intensity efforts. The inspiratory muscles, including the diaphragm and intercostals, support core stability and efficient oxygen delivery during dynamic movements [15]. In sports like soccer, inspiratory muscle performance, assessed via measures like maximal inspiratory pressure (MIP), has been correlated with lower extremity strength and endurance [16]. In baseball, pitching involves rapid trunk rotation and force transfer, increasing respiratory demands [17]. Inspiratory muscle fatigue may reduce pitching velocity and alter mechanics, potentially elevating injury risk [18]. Preliminary evidence suggests that inspiratory muscle fatigue may contribute to decreased pitching velocity and altered mechanics, potentially increasing injury risk [18]. The relationship between inspiratory performance (IP) and glenohumeral motion is underexplored, particularly in elite-level baseball, where unique biomechanical demands may amplify these interactions.

This study investigated the relationships between inspiratory performance (IP) and glenohumeral rotation in Division 1 Collegiate baseball players (D1CBP). We hypothesized that stronger IP is associated with reduced external rotation, potentially due to enhanced core stability limiting excessive shoulder motion. This study addresses a research gap, as prior studies have not integrated respiratory and shoulder mechanics in this population. The findings could inform training strategies, such as inspiratory muscle training (IMT), to optimize performance and reduce injury risk.

## 2. Materials and Methods

### 2.1. TIRE Testing

The MIP, as shown in Figure 1, indicates the highest pressure generated by an individual during the first second of an inspiratory breath [5]. This metric captures the peak strength of the inspiratory muscles. In contrast, the SMIP quantifies the cumulative pressure produced over the entire duration of a sustained inhalation, providing insights into the strength, endurance, and work capacity of these muscles during a single breath [5]. In Figure 1, MIP is represented by the initial peak of the inspiratory breath, while SMIP is depicted as the area under the pressure curve, highlighted in red.

### 2.2. Participants

Thirty D1CBP from the same team participated in this study, with the physical characteristics shown in Table 1. Of the 30 participants, 14 were pitchers (7 left-handed and 7 right-handed), 11 were right-handed infielders, and 5 were outfielders (2 left-handed and 3 right-handed).

The grouped bar plot (Figure 2) displays the count of each academic year (freshman, sophomore, junior, senior) for each position group (pitchers, infielders, outfielders) for comparison across positions.

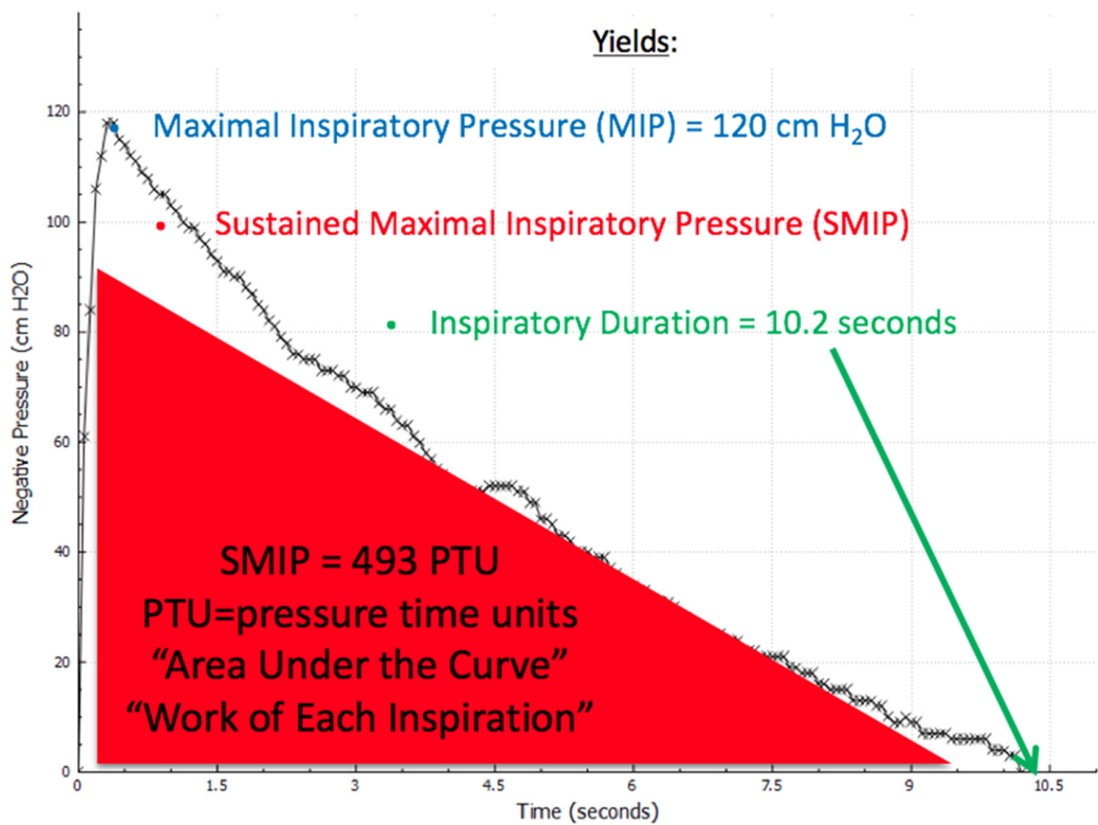

**Figure 1.** The pressure-time curve was produced using the RT2 device during the Test of Incremental Respiratory Endurance (TIRE) protocol. Maximal Inspiratory Pressure (MIP), measured in cm $H_2O$, reflects inspiratory muscle strength, with higher values indicating greater strength. Sustained Maximal Inspiratory Pressure (SMIP), quantified in Pressure Time Units (PTU), represents the inspiratory pressure sustained throughout inhalation, with higher values corresponding to increased single-breath work capacity. Inspiratory Duration, measured in seconds, indicates inspiratory endurance, with longer durations associated with greater endurance.

**Table 1.** Anthropometric Characteristics of the Baseball Players.

| Variables | Mean ± SD |
|---|---|
| Age (years) | 20 ± 1.5 |
| Height (cm) | 184.5 ± 5.9 |
| Weight (kg) | 89.1 ± 8.1 |
| Body Mass Index | 26.1 ± 1.7 |

A power analysis was conducted to determine the sample size necessary to detect a significant relationship between inspiratory performance and glenohumeral rotation. Based on an estimated large effect size (r = 0.5), a significance level of $p < 0.05$, and power (1) of 0.80, the minimum sample size, based on prior studies required was determined to be 29 [16]. Thus, the current sample of 30 participants meets the threshold for detecting meaningful relationships within this specific population.

Inclusion criteria for participation included being a D1CBP and being free from respiratory or musculoskeletal conditions that could impair normal function. Exclusion criteria included recent injuries (within the past 6 months), or respiratory illnesses. Participants were recruited through convenience sampling from a single Division I collegiate baseball team.

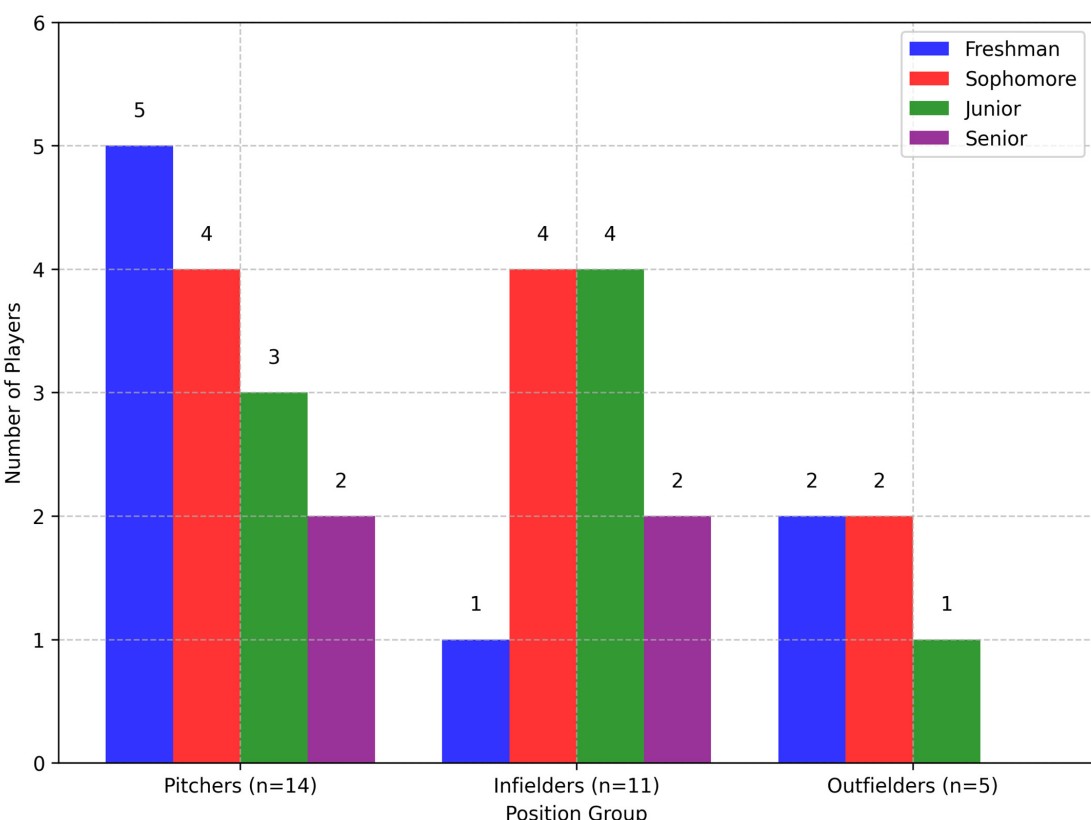

**Figure 2.** Distribution of academic years by position. This figure shows the distribution of players across various positions (pitchers, infielders, outfielders) and their academic years.

### 2.3. Procedures

Participants provided informed consent prior to study participation, and the research adhered to the principles of the Declaration of Helsinki. The study protocol received approval from the University's institutional ethics committee, and no external funding was obtained for this research. Testing occurred during the pre-season in a controlled environment (22–24 °C, 50–60% humidity). Participants completed a standardized 10 min warm-up (light stretching and mobility exercises) before testing to minimize variability.

#### 2.3.1. Shoulder Mobility Testing

Glenohumeral internal and external rotation (RGHIR, RGHER, LGHIR, LGHER) were measured using a handheld goniometer (Fabrication Enterprises Inc., White Plains, NY, USA) by a single trained assessor (a physical therapist who is a board-certified sports clinical specialist). All participants had prior experience with goniometric testing as part of their routine pre-participation performance evaluations. During testing, participants were supine with their shoulder abducted to 90d, the forearm in neutral, and the length of the humerus on the test side supported on a plinth to maintain proper alignment. A belt stabilized the thorax to prevent compensatory trunk rotation. The axis of rotation was aligned with the olecranon process of the ulna, the stationary arm was perpendicular to the floor, and the movement arm was in line with the ulnar side of the forearm from the axis point to the ulnar styloid process [2]. Participants were instructed to relax as the tester rotated their shoulder through their full range of motion. Total rotational shoulder motion (TRM) was calculated by the summation of internal and external rotation motion [2] Thus, the following measures were obtained, including right and left glenohumeral internal and external rotation (RGHIR, RGHER, LGHIR, and LGHER, respectively) as well as right and left total rotational motion (RTRM and LTRM, respectively).

### 2.3.2. Test of Incremental Respiratory Endurance (TIRE)

The TIRE protocol used the RT2 device (DeVilbiss Healthcare Ltd., Wollaston, UK), calibrated before each session per manufacturer guidelines by a single trained assessor (a physical therapist who specializes in cardiopulmonary physical therapy with over 30 years of experience in the field). Participants were familiarized with the protocol via a practice trial. They exhaled fully, then inhaled forcefully and sustained the effort as long as possible, prompted by "deep, long, and hard" cues. Three to five trials were conducted in a seated position, with the highest MIP and SMIP trial selected (prioritizing MIP). Verbal encouragement and real-time biofeedback were provided via TIRE software (v2.52). SMIP was measured in Pressure-Time Units (PTU), representing the area under the inspiratory pressure-time curve (Figure 1).

### 2.4. Statistical Analyses

All statistical analyses were conducted using SPSS v28, with the significance level set at $p < 0.05$. Descriptive statistics were calculated for all variables, including means and standard deviations. The Shapiro–Wilk test was used to assess normality for continuous variables. Since all variables met the assumption of normality ($p > 0.05$), parametric tests were applied. A one-way analysis of variance (ANOVA) was used to examine differences among position groups (pitchers, infielders, outfielders) for anthropometric, biomechanical, and inspiratory performance variables. When significant main effects were observed, post hoc Tukey's HSD tests were conducted to determine pairwise group differences. Effect sizes for between-group comparisons were calculated using Cohen's d, interpreted as small (0.2), medium (0.5), or large (0.8). A large effect size would indicate clinically significant differences in inspiratory muscle endurance between player positions.

Pearson product-moment correlation coefficients were selected for their reliability with normally distributed data, to evaluate relationships between sustained maximal inspiratory pressure (SMIP), measured in pressure-time units ([PTU]), and relevant anthropometric and biomechanical variables within the full cohort and specific subgroups. Bonferroni correction was applied to control for multiple comparisons in correlation analyses, with the adjusted alpha level set at 0.0083. All data were normally distributed based on Shapiro–Wilk tests ($p > 0.05$), permitting the use of parametric statistical tests.

## 3. Results

### 3.1. Descriptive Statistics

Inspiratory and glenohumeral mobility characteristics are shown in Table 2.

**Table 2.** Characteristics of Inspiratory Performance and Glenohumeral Mobility.

| | |
|---|---|
| Maximal Inspiratory Pressure (cm $H_2O$) | $113.4 \pm 27$ |
| Sustained Maximal Inspiratory Pressure (PTU) | $722.1 \pm 309.9$ |
| Inspiratory Duration (s) | $11.6 \pm 3.5$ |
| Right Glenohumeral Internal Rotation (deg) | $54 \pm 13.5$ |
| Right Glenohumeral External Rotation (deg) | $103.8 \pm 12$ |
| Right Glenohumeral Total Rotational Motion (deg) | $157.8 \pm 15.9$ |
| Left Glenohumeral Internal Rotation (deg) | $59.8 \pm 13.9$ |
| Left Glenohumeral External Rotation (deg) | $98.2 \pm 13.3$ |
| Left Glenohumeral Total Rotational Motion (deg) | $158.0 \pm 18.3$ |

### 3.2. Between-Group Comparison

Inspiratory performance across three position groups can be found in Table 3. Outfielders exhibited the highest mean maximal inspiratory pressure ($135 \pm 18$ cm $H_2O$)

and a moderate SMIP (693 ± 153 PTU), while pitchers had the highest mean SMIP (798 ± 403 PTU) and longest inspiratory duration (12.4 ± 4.2 s), with infielders showing the lowest values across all measures (MIP: 108 ± 30 cm $H_2O$, SMIP: 639 ± 206 PTU, inspiratory duration [ID]: 11.1 ± 2.6 s).

**Table 3.** Inspiratory Performance by Position.

|  | MIP (cm $H_2O$) | SMIP (PTU) | ID (s) |
|---|---|---|---|
| Pitchers (n = 14) | 110 ± 25 | 798 ± 403 | 12.4 ± 4.2 |
| Infielders (n = 11) | 108 ± 30 | 639 ± 206 | 11.1 ± 2.6 |
| Outfielders (n = 5) | 135 ± 18 | 693 ± 153 | 10.7 ± 3.1 |

One-way ANOVA revealed significant differences between position groups on SMIP ($F_{(2,27)}$ = 6.12, *p* = 0.006), RGHER ($F_{(2,27)}$ = 7.34, *p* = 0.003), and RTRM ($F_{(2,27)}$ = 6.85, *p* = 0.004) (Figure 3). Post hoc Tukey tests indicated that pitchers had significantly higher SMIP (mean: 784.6 ± 315.2 PTU) compared to infielders (mean: 672.3 ± 298.7 PTU, *p* = 0.008, Cohen's d = 0.91) and outfielders (mean: 650.4 ± 304.1 PTU, *p* = 0.02, Cohen's d = 1.02). Pitchers also exhibited greater RGHER (mean: 108.9 ± 11.5 degrees) and RTRM (mean: 163.2 ± 15.1 degrees) compared to infielders (RGHER: 99.1 ± 11.2 degrees, *p* = 0.004, Cohen's d = 1.05; RTRM: 152.4 ± 15.4 degrees, *p* = 0.006, Cohen's d = 0.98) and outfielders (RGHER: 98.4 ± 11.8 degrees, *p* = 0.01, Cohen's d = 1.10; RTRM: 151.8 ± 15.9 degrees, *p* = 0.02, Cohen's d = 1.03). No significant differences were found between infielders and outfielders for SMIP, RGHER, or RTRM (*p* > 0.05). Additionally, no significant differences were observed for maximal inspiratory pressure (MIP), inspiratory duration, right glenohumeral internal rotation (RGHIR), left glenohumeral external rotation (LGHER), or left total rotational motion (LTRM) across position groups (*p* > 0.05).

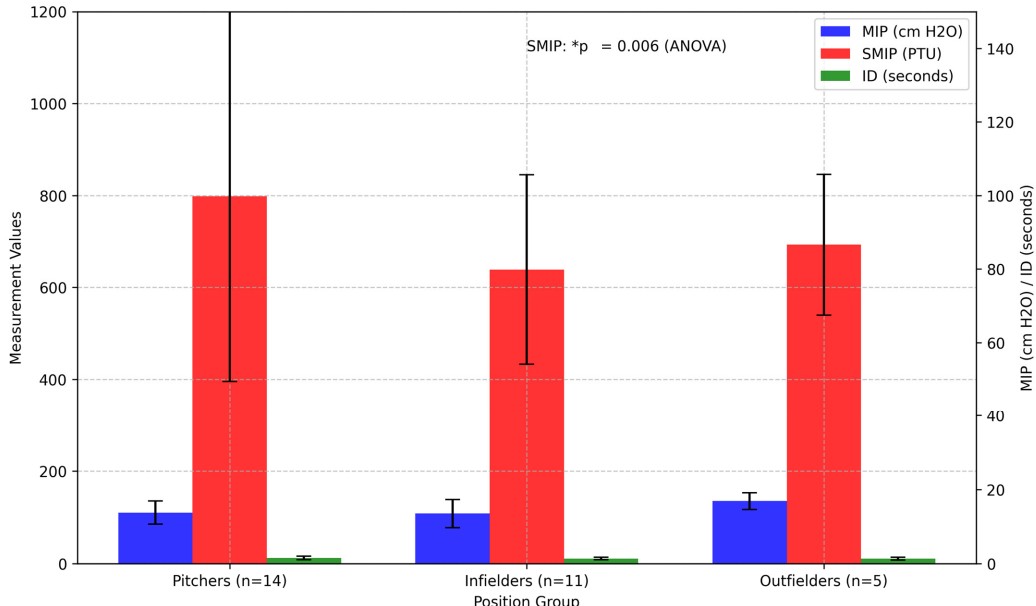

**Figure 3.** Inspiratory performance by position in D1CPBs. The bar plot shows the MIP, SMIP, and ID for the three position groups. SMIP showed significant differences (* *p* = 0.006).

### 3.3. Correlation Analysis

Significant relationships were found between SMIP and both anthropometric and biomechanical variables for the entire cohort (Table 4). SMIP was positively correlated with height (r = 0.38, *p* = 0.008) and weight (r = 0.42, *p* = 0.003), suggesting that taller and heavier players exhibited greater inspiratory muscle endurance and single-breath work capacity.

Conversely, SMIP was negatively correlated with RTRM (r = −0.41, *p* = 0.004), suggesting that greater total rotational motion in the right shoulder was associated with lower SMIP values.

**Table 4.** Correlation Coefficients Between SMIP and Other Variables.

| Variable | Full Cohort (n = 30) | Pitchers Only (n = 14) | RHP (n = 7) | LHP (n = 7) |
|---|---|---|---|---|
| Height | r = 0.38, *p* = 0.008 * | - | - | - |
| Weight | r = 0.42, *p* = 0.003 * | - | - | - |
| RTRM | r = −0.41, *p* = 0.004 * | R = −0.56, *p* = 0.005 * | - | - |
| RGHER | - | - | R = −0.88, *p* < 0.001 * | ns |
| LGHER | - | - | ns | R = −0.82, *p* = 0.04 * |

* *p* < 0.0083 after Bonferroni correction; - indicates non-significant results (*p* > 0.0083) or not applicable; ns indicates non-significant results (*p* > 0.0083). RTRM = right total rotational motion; RGHER = right glenohumeral external rotation; LGHER = left glenohumeral external rotation; RHP = Right-Handed Pitchers; LHP = Left-Handed Pitchers.

The negative correlations between SMIP and glenohumeral motion metrics (RTRM, RGHER, LGHER) for pitchers are visualized in Figure 4. Among pitchers (n = 14), SMIP was significantly negatively correlated with RTRM (r = −0.56, *p* = 0.005) and LTRM (r = −0.61, *p* = 0.002). For right-handed pitchers (n = 7), SMIP showed a strong negative correlation with RGHER (r = −0.88, 95% CI: −0.83 to −0.93, *p* < 0.001). For left-handed pitchers (n = 7), SMIP was negatively correlated with LGHER (r = −0.82, *p* = 0.04). These findings indicate that higher SMIP values were associated with reduced external rotation in the dominant throwing arm. No significant correlations were found between SMIP and LGHER in right-handed pitchers or between SMIP and RGHER in left-handed pitchers (*p* > 0.05). All significant correlations remained significant after Bonferroni correction for multiple comparisons (adjusted α = 0.0083).

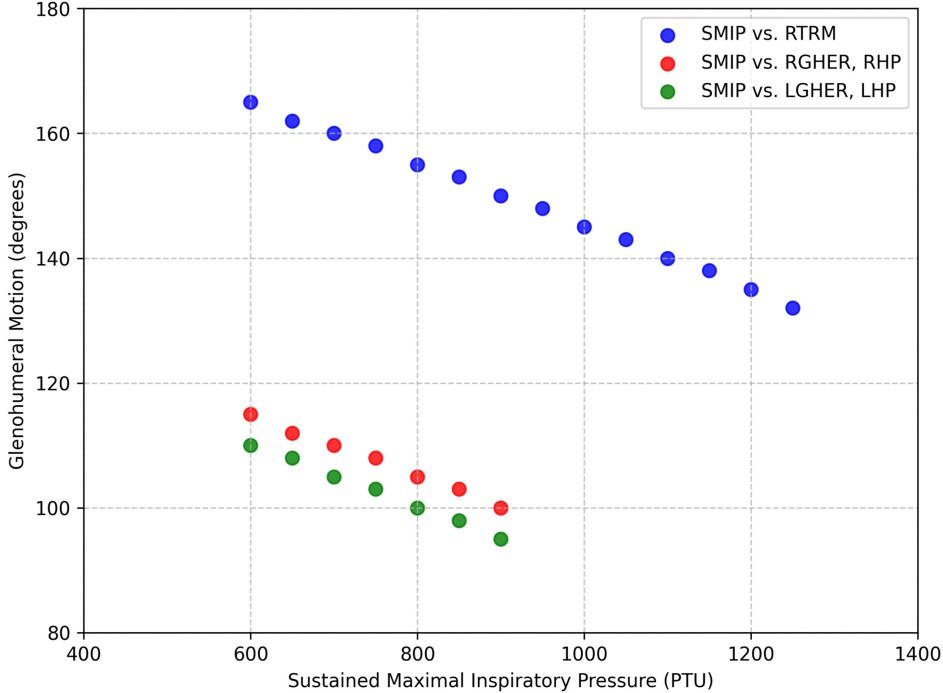

**Figure 4.** Negative correlations between SMIP and glenohumeral motion in pitchers. This scatter plot illustrates the negative correlations between SMIP and glenohumeral motion metrics (RTRM, RGHER, LGHER) for pitchers, as reported: RTRM (r = −0.56), LGHER (r = −0.82 for LHP), and RGHER (r = −0.88 for RHP).

## 4. Discussion

The findings of this study reveal significant differences in inspiratory muscle function and glenohumeral motion between position groups among Division I collegiate baseball players, alongside notable correlations between sustained maximal inspiratory pressure (SMIP) and both anthropometric and biomechanical variables. These results provide insights into the interplay between respiratory muscle performance and shoulder mechanics in elite pitchers, supporting our hypothesis that inspiratory muscle function may influence glenohumeral range of motion (ROM) and, by extension, pitching performance and injury risk.

### 4.1. Positional Differences

Significant differences in SMIP and glenohumeral motion metrics (e.g., RTRM, LTRM, RGHER, and LGHER] between position groups align with prior research indicating positional specialization in baseball. Pitchers, who experience repetitive, high-intensity throwing, likely develop distinct biomechanical adaptations compared to position players, as evidenced by studies on shoulder ROM and muscle recruitment patterns [1,2]. For instance, Wilk et al. [2] reported that pitchers exhibit greater external rotation and reduced internal rotation in the throwing arm, a pattern consistent with GIRD. Our findings extend this by suggesting that inspiratory muscle performance, as measured by SMIP, also varies by position, potentially reflecting the greater respiratory demands of pitching, which involves rapid trunk rotation and force transfer [17]. These positional differences demonstrate the need for tailored training programs that address both respiratory and shoulder mechanics specific to player roles.

### 4.2. Correlations with Anthropometrics

The positive correlations between SMIP and anthropometric measures (height and weight) in the entire cohort suggest that larger body size may confer advantages in inspiratory muscle capacity, consistent with studies in other athletic populations [15]. Taller and heavier athletes are likely to have greater diaphragmatic descent and greater lean muscle mass, contributing to stronger primary and accessory muscles of breathing, resulting in higher SMIP values and greater lung volumes. These factors may be particularly relevant for pitchers, who require enhanced levels of core stability to support the kinetic chain during throwing [19]. However, the negative correlation between SMIP and RTRM ($r = -0.41$, $p < 0.05$) across all players, and stronger negative correlations with RTRM and LTRM in pitchers ($r = -0.56$ to $-0.61$, $p < 0.05$), indicate an interplay between inspiratory muscle endurance/work and shoulder mobility. These findings suggest that greater inspiratory muscle performance may be associated with reduced total rotational motion, potentially due to increased core stiffness that limits excessive shoulder rotation [18]. This hypothesis contrasts with studies suggesting that enhanced respiratory muscle function improves overall athletic performance without restricting joint mobility [3,20]. The discrepancy may reflect sport-specific adaptations in baseball, where excessive glenohumeral rotation is a known risk factor for shoulder injuries [10].

### 4.3. Correlations with ROM

Particularly striking are the strong negative correlations between SMIP and RGHER in right-handed pitchers ($r = -0.83$ to $-0.93$, $p < 0.05$) and LGHER in left-handed pitchers ($r = -0.82$, $p = 0.04$). These findings suggest that pitchers with greater inspiratory muscle endurance/work may exhibit reduced external rotation in their throwing arm, potentially as a protective mechanism against excessive ROM. Previous studies have linked increased external rotation to higher pitching velocity but also to greater injury risk, such as rotator

cuff tears and internal impingement [6,9]. The negative correlation between SMIP and RGHER/LGHER may indicate that stronger inspiratory muscles enhance core stability, thereby constraining shoulder motion to prevent pathological over-rotation [21]. This interpretation aligns with biomechanical models of the kinetic chain, where trunk stability modulates force transmission to the upper extremity [12]. However, it diverges from studies in other sports, such as soccer, where inspiratory muscle training improved performance without altering joint mechanics [16]. These sport-specific differences show the unique biomechanical demands of pitching and suggest that inspiratory muscle function may play a dual role in enhancing performance while mitigating injury risk.

*4.4. Practical Implications*

The implications of these findings are significant for both performance optimization and injury prevention in collegiate baseball. The negative correlations between SMIP and glenohumeral motion metrics suggest that inspiratory muscle training could be integrated into conditioning programs to enhance core stability and potentially reduce excessive shoulder rotation, a known contributor to GIRD and related injuries [7]. However, the optimal balance between inspiratory muscle strength and shoulder mobility remains unclear, as excessive core stiffness could theoretically limit pitching efficiency [14]. Coaches and athletic trainers should consider individualized assessments of SMIP and glenohumeral ROM to tailor interventions, particularly for pitchers who exhibit extreme external rotation values.

Fatigue from repetitive throwing in baseball pitchers disrupts pitching biomechanics, performance, and increases injury risk, with inspiratory muscle training (IMT) offering a multifaceted solution by enhancing core stability and supporting peripheral muscle function through the metaboreflex. Repetitive pitching impairs the kinetic chain, reducing shoulder and elbow proprioception, decreasing ball velocity and accuracy, and causing muscle soreness in the forearm flexors, anterior deltoid, and upper trapezius [22]. These changes, combined with altered hip and torso kinematics (e.g., reduced stride length and torso rotation), heighten stress on the throwing arm, elevating the risk of overuse injuries like rotator cuff tears and elbow ligament damage [22–24]. Higher external workloads for fastballs, as measured by tri-axial accelerometers, further exacerbate fatigue, strengthening the need for targeted interventions [24]. IMT strengthens the diaphragm and co-activates scapular stabilizers like the serratus anterior, enhancing core stability and limiting excessive glenohumeral rotation, which may reduce injury risk.

IMT also mitigates fatigue through the respiratory metaboreflex, a mechanism where inspiratory muscle fatigue triggers sympathetic activation, reducing blood flow to peripheral muscles and accelerating limb fatigue [24]. By improving inspiratory muscle performance, IMT delays the onset of this metaboreflex, enhancing oxygen delivery and metabolic efficiency to working muscles like those in the shoulder, elbow, and lower body, thereby sustaining explosiveness and pitching velocity [22–24]. Integrating IMT with strength training for hip and torso muscles, proprioceptive exercises to improve joint position sense, and workload monitoring via wearable technology can further address fatigue-induced declines in accuracy and soreness while preventing overuse [22–24]. These strategies utilize IMT's potential to optimize performance and protect against fatigue-related injuries in baseball pitchers by supporting both central stability and peripheral muscle function [22–24].

In a broader context, these results contribute to the growing recognition of the respiratory system's role in athletic performance, extending beyond the traditional focus on the cardiovascular and musculoskeletal systems [25–29]. The interplay between inspiratory muscle function and glenohumeral motion suggests that whole-body training approaches,

integrating respiratory, core, and upper extremity conditioning, may enhance pitching performance and longevity. Hodges and Gandevia's findings support the argument that diaphragm contraction is related to trunk control during repeated upper limb and ribcage movements, indicating that activity of human phrenic motor neurons are organized such that it contributes to both posture and respiration [30]. This study also highlights the importance of considering positional differences in training and rehabilitation protocols, as pitchers face unique biomechanical and physiological demands compared to other players.

*4.5. Limitations*

This study has several limitations that future research should address. The single-team sample and small outfielder group (n = 5) limit the generalizability of findings, particularly across different baseball populations, such as professional or younger athletes. The cross-sectional design prevents determining causality, and test–retest reliability was not assessed. Potential measurement errors in TIRE and goniometry, despite calibration and assessor reliability efforts, may also affect results. To build on these findings, longitudinal studies are needed to explore whether targeted inspiratory muscle training, which improves SMIP, directly impacts glenohumeral motion and injury rates. Investigating the mechanistic link between inspiratory muscle performance and shoulder mechanics, potentially using electromyography or motion capture, could clarify how core stability influences throwing biomechanics [31–36]. Additionally, examining other respiratory parameters, such as expiratory muscle strength, could provide a more comprehensive understanding of respiratory contributions to pitching and throwing performance [30–35]. Expanding the sample to include diverse athlete groups would help determine if these findings are specific to Division I collegiate players or apply more broadly.

## 5. Conclusions

Pitchers exhibit higher SMIP and greater glenohumeral external rotation than other positions, with SMIP negatively correlated with shoulder ROM, particularly in pitchers. This study pioneers the integration of inspiratory and shoulder mechanics in D1CBP, addressing a research gap. Incorporate IMT into training to enhance core stability and potentially reduce injury risk by limiting excessive shoulder rotation. Further longitudinal studies with larger, more diverse samples and EMG/biomechanical modeling are needed to validate findings and explore causality.

**Author Contributions:** Conceptualization, L.A.F., L.P.C. and M.A.R.; methodology, L.A.F., M.I.C., J.J.R., M.A.R. and L.P.C.; formal analysis, L.P.C.; investigation, L.A.F. and L.P.C.; resources, L.D.K. and J.J.R.; data curation, L.P.C.; writing—original draft preparation, L.A.F. and L.P.C.; writing—review and editing, M.I.C. and M.A.R.; visualization, L.P.C.; supervision, L.A.F. and L.P.C.; project administration, L.A.F. and L.P.C. All authors have read and agreed to the published version of the manuscript.

**Funding:** This research received no external funding.

**Institutional Review Board Statement:** The study was conducted in accordance with the Declaration of Helsinki and approved by the Institutional Review Board of the University of Miami (protocol code 2012034 and 1 June 2018) for studies involving humans.

**Informed Consent Statement:** Informed consent was obtained from all subjects involved in the study.

**Data Availability Statement:** The datasets presented in this article are not readily available because the data are part of an ongoing study.

**Conflicts of Interest:** The authors declare no conflicts of interest.

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
