# Peer review of "Assessment of Inspiratory Muscle Function and Glenohumeral Motion in the Throwing Arm of Division I Collegiate Baseball Players"

_applsci, doi:10.3390/app15168815_

Round 1

Reviewer 1 Report

Comments and Suggestions for Authors

The article presents a concern about aspects of physiological nature that can influence individual performance in sports.
It was chosen to study the respiratory process and its influence on the unitary movements for baseball athletes.
The article presents the introduction, method, results, discussions, conclusions and references.
The method is presented briefly. The way of selecting the lot of subjects and the procedure for applying tests is presented here.
The results of the results show for the following parameters that were the results recorded by the lot of subjects.
The part of the statistical analysis is presented.
In the part of discussions, various correlations between registered parameters and their influence in the respiratory process are presented. It is too little presented and discussed with the sports performance part of the mechanisms studied.
The conclusions are in agreement with the researched subject. The conclusions open perspectives for potential future research on this subject.
The article is well done in terms of form.
From the point of view of the scientific content, it tries to make a contribution to understanding the physiological process that could influence an increased level of throwing performance.
The article can be accepted for publication in presented form.

Author Response

Thank you for your thoughtful consideration, dedicated effort, and valuable time in reviewing this manuscript submission. I am grateful for your positive feedback and support for publishing this work without further modifications. I acknowledge that several other reviewers have suggested additional revisions to enhance the clarity, depth, and presentation of the study's results. We are committed to addressing these requests diligently and will incorporate the necessary revisions to strengthen the manuscript.

Reviewer 2 Report

Comments and Suggestions for Authors

This study explores the relationship between inspiratory muscle function (IP) and glenohumeral motion in the throwing arm of Division I collegiate baseball players. The topic is innovative, filling a research gap in the biomechanical association between respiratory muscles and shoulder mechanics in baseball. The study design is rational, the data are comprehensive, and the statistical methods are rigorous. The conclusions have potential practical implications for sports training (e.g., integrating respiratory training with shoulder rehabilitation). However, some methodological details need refinement. ​

​Sample Size Justification​​
The study mentions that sample size was calculated based on an effect size of r = 0.5, but the source of this effect size (e.g., pilot study, previous literature) is unclear.
​​Recommendation​​: Provide a reference or rationale for selecting r = 0.5 (e.g., "Based on a meta-analysis of inspiratory muscle training effects in athletes [citation]").
TIRE Test Protocol​​
While the TIRE test procedure is described (Figure 1 and text), details about the testing environment (e.g., room temperature, participant warm-up status) are missing.
​​Recommendation​​: Specify whether environmental conditions (e.g., standardized room temperature, rest period before testing) were controlled to minimize variability in MIP/SMIP measurements.
Shoulder ROM Measurement​​
The description of shoulder ROM assessment ("supine position, 90° abduction") lacks detail on whether trunk stabilization (e.g., belt fixation to prevent compensatory rotation) was used.
​​Recommendation​​: Clarify whether measures were taken to prevent trunk movement (e.g., "A belt was used to stabilize the thorax and minimize compensatory rotation during measurements [reference]").
Results Presentation Could Be Improved​​
Table Clarity​​
Table 3 (IP data by position) reports SMIP in "PTU" (Pressure-Time Units), but this unit is not defined in the table footnote (though mentioned in the abstract).
​​Recommendation​​: Define PTU in the table footnote or main text (e.g., "PTU: Pressure-Time Units, representing the area under the inspiratory pressure-time curve").
Effect Size Interpretation​​
Cohen’s d values (e.g., d = 0.91 for SMIP differences between pitchers and infielders) are reported without interpretation of their practical significance.
​​Recommendation​​: Discuss effect sizes in context (e.g., "d = 0.91 indicates a large effect, suggesting clinically meaningful differences in inspiratory muscle endurance between positions [citation]").
Discussion Requires Deeper Analysis​​
Mechanistic Insights​​
The hypothesis that SMIP relates to glenohumeral rotation via core stability (e.g., "SMIP-GH external rotation negative correlation due to stiffness limiting excessive rotation") lacks neurophysiological support.
​​Recommendation​​: Integrate literature on respiratory-diaphragmatic and scapular stabilizer (e.g., serratus anterior) co-activation (e.g., Hodges et al. on respiratory-motor control [citation]).
Limitations and Future Directions​​
​​Single-center bias​​: The study’s generalizability is limited by a single-team sample.
​​Lack of validation​​: No mention of test-retest reliability or independent replication.
​​Recommendation​​:
Acknowledge limitations (e.g., "Findings may not generalize to other populations due to the single-team design").
Propose future work (e.g., "Longitudinal studies are needed to determine if inspiratory muscle training alters shoulder ROM and injury risk").

Author Response

I have included an attached document outlining the changes made in response to your feedback. We hope the changes to the submission are an upgrade to the original draft. We would like to thank you for your time and consideration.

Reviewer 3 Report

Comments and Suggestions for Authors

Comment 1:

The abstract is comprehensive and provides a clear overview of the study.

Suggestions to emphasize the main findings and their practical implications more effectively.:

1.Consider restructuring the abstract to follow a clearer format: (1) Background, (2) Aim, (3) Methods, (4) Results, (5) Conclusions and Implications.

2.Provide specific numerical outcomes (e.g., correlation coefficients, p-values) to support the key findings.

3.The last sentence is a bit technical. Simplify or clarify the explanation regarding the overstretched accessory inspiratory muscles.

Comment 2:

In the introduction, the logical flow can be improved to clarify how the two main physiological domains—inspiratory muscle performance and shoulder biomechanics—interrelate.

Please, strengthen the transition between the background on GIRD and the role of inspiratory muscles. Currently, these two topics are discussed in parallel but could be better integrated. Clarify why exploring this relationship in collegiate pitchers specifically is novel and important. Reduce repetition—some ideas (e.g., throwing adaptations, injury risk) appear multiple times with only slight variations. Improve flow by using clear linking sentences that explain how previous findings lead to the current research question.

Comment 3:

In the Methods section, further elaboration and clarification in certain parts would improve reproducibility and transparency.

1.TIRE protocol - indicate whether the device was calibrated before testing and whether participants were familiarized with the protocol before data collection.

2.Goniometry procedure - while references are cited, please specify the intra- and inter-rater reliability of these measures if available (or mention if the same assessor conducted all measurements).

3.Sample size - although a power analysis was conducted, explicitly state effect size assumptions and statistical software used.

4.statistical analysis - the description is thorough; however, it would be helpful to explain why Pearson correlations were chosen and how normality assumptions were tested and confirmed.

Comments 4:

The Results section is rich in data, and findings are statistically supported. Still, the narrative could be improved for clarity and emphasis.

Structure the section more clearly by separating the results into (1) Descriptive statistics, (2) Between-group comparisons, and (3) Correlation analysis. Highlight key outcomes by bolding or italicizing main statistical values (in the typeset version). Clarify certain statistics—e.g., it is unclear whether SMIP differences are clinically significant in addition to statistically significant.

In Table 4, provide explanations for RHP and LHP acronyms in the caption. Also, clearly indicate whether correlation values refer to MIP, SMIP, or ID.

Comment 5:

The Discussion would benefit from better organization and more concise phrasing. Begin by restating the main finding more directly and succinctly. Break down the Discussion into clear subsections: e.g., (1) Positional Differences, (2) Correlations with Anthropometrics, (3) Correlations with ROM, (4) Practical Implications, and (5) Mechanistic Interpretation. Clarify the hypothesized mechanism for the negative correlation between SMIP and external rotation. Consider using a diagram or conceptual model in the published version. Cite more literature specific to baseball pitching mechanics and inspiratory fatigue, especially in the context of training applications. Be cautious in interpreting causality; the cross-sectional design only allows for associations, not directional conclusions.

Comment 6:

Limitations are addressed, but could be expanded and more critically discussed. You can mention the cross-sectional design explicitly as a limitation in establishing causality. The homogeneity of the sample (one Division I team) limits generalizability—suggest expanding to other levels (e.g., high school, pro). Acknowledge potential measurement error or learning effect in the TIRE protocol or goniometry, even if mitigated. Suggest integrating EMG or biomechanical modeling in future studies to better understand the observed relationships.

Comment 7:

The conclusion accurately summarizes the main findings, but could be more impactful if aligned with practical recommendations. Consider framing the conclusion in terms of:

  1. Key finding
  2. Contribution to the field
  3. Practical recommendation
  4. Direction for future research.

Emphasize how the findings might inform preventive strategies or training protocols, especially for pitchers. Avoid speculative statements; keep conclusions grounded in data and realistic in scope.

Author Response

(The authors gave the same response as above.)

Reviewer 4 Report

Comments and Suggestions for Authors

Dear Authors,

Below are my observations. I hope it helps

AIM
The aim proposed in L13-14 (This study examined the relationships of inspiratory performance (IP) to characteristics of Division 1 Collegiate baseball players", the one proposed in L39-41 ("This study investigates the interplay between..... in a collegiate population") and the one proposed in L68-70 ("The aim of this study was to investigate... collegiate baseball players") are different in terms of wording and even in basic idea. I recommend that authors establish a clear aim, which they should state everywhere in the manuscript where this aspect is requested, although once is sufficient, and then taken up in the abstract.

INTRODUCTION
The position of figure 1 is not an ok one, it appears in the text before it is discussed/trimetered. I think its place is not in the Introduction, but in Materials and Methods.

METHODS
L89-97: Section "2.1 TIRE Testing" does not present how the measurement was performed, but describes theoretically what it represents. NO details are specified about how the maximal inspiratory pressure (MIP) was measured in the subjects: location, time, personnel who evaluated, etc. etc. the procedure is not specified here. From what I have observed the procedure is described in 2.3. which requires that figure 1 and the theoretical explanation be included either in the Introduction, as part of a paragraph, or better in 2.3. as part of the Procedures section. I do not see the point of section 2.1. as it is now and placed in the manuscript at L89-97.

In the section dedicated to the inclusion criteria but also in table no. 1, the age / experience in practicing performance sports and the age in practicing baseball should be specified. As a criterion for presenting the characteristics of the group of subjects, nothing is specified about how many years the subjects have been doing performance sports (relevant index for respiratory power) and how many years they have been practicing baseball (relevant index for the demand on the joint targeted in this study). I believe that these two aspects are essential!!

L114: “met the athletic requirements” ??? what would these “athletic requirements” be if the authors bring them into discussion, because the wording is very vague. Nothing is specified until L114 about any criteria that would reach “athletic requirements”!!! I think it is a wording that has no connection with Materials and Methods. If it does, then the authors should argue!

RESULTS
L168-170 repeat what has already been presented in a previous section. It is not justified to include the same data about the subjects here.
For the significant differences I recommend building a diagram, it is much more suggestive for the readers, along with the description of the results in the text.

DISCUSSION
Ok, compared to what the Results section has exposed

CONCLUSIONS
The first conclusion, as formulated, is not supported by the size of the group of subjects!! If the playing positions are brought into discussion, you cannot generalize by comparing 14, 11 and 5, 5 being insignificant to propose such a conclusion. I recommend a reformulation.
This conclusion “These findings advocate for a holistic approach to training that considers the interplay between respiratory and shoulder function, with potential applications for enhancing performance and reducing injury risk” is not subordinate to the title and is not the subject of the scope of this manuscript. It is much too general, not being supported by the data in Results. I recommend to the authors that this section, dedicated to Conclusions, be reported exclusively to what essentially resulted from the experiment, from the data presented, and not to present personal assumptions, unsupported by the data.

OTHER ASPECTS
- the study’s limitations are not presented
- I recommend an emphasis on the practical applications of this research
- almost 55% of the references are older than 10 years, some going towards 20 years. I do not claim that it would be bad, however, I also recommend the inclusion of newer studies from baseball or other sports disciplines in which they are analyzed throws in relation to respiratory function
- to achieve reproducibility criteria, authors should more clearly detail the Procedures and tests section. Provide details on how subjects were evaluated, at what times of day, in what period of the competitive year (training or competition, given the differences in tone, accumulated fatigue, etc.), whether the joint assessment was done directly, without training movements or were some training movements, joint mobility, performed before evaluating rotations, etc. etc.

-I would like 1-2 diagrams in the Results section, which would suggestively present the differences in the data obtained, which would help the readers much more to understand

Author Response

(The authors gave the same response as above.)

Round 2

Reviewer 2 Report

Comments and Suggestions for Authors

none

Author Response

Thank you for your time and consideration with the updated submission. Your efforts are very much appreciated. 

Reviewer 3 Report

Comments and Suggestions for Authors

I have reviewed the revised version of the manuscript entitled “Assessment of Inspiratory Muscle Function and Glenohumeral Motion in the Throwing Arm of Division I Collegiate Baseball Players.” I appreciate the authors’ efforts in addressing the comments provided during the first round of peer review. Overall, the manuscript has significantly improved in clarity, structure, and scientific rigor.

Minor suggestions for further improvement:

  • in the Abstract, the sentence “The link between IP and GHER likely comes from GHER and overextended GHIR muscles, which help with breathing” may still be improved slightly for clarity. Consider rephrasing to: “This association may be explained by the involvement of overstretched internal rotators, which act as accessory inspiratory muscles.”
  • for final publication, ensure all figures (especially Figures 3 and 4) are accompanied by fully descriptive captions, including sample sizes and clear axes labels.

I am satisfied with the revisions made and recommend the manuscript for publication after minor editorial adjustments.

Author Response

Thank you for your time and consideration with the updated submission. Your efforts are very much appreciated. I will work on refining that sentence in the abstract and providing greater detail in Figures 3 and 4.